# Using a Machine Learning Approach to Predict Snakebite Envenoming Outcomes Among Patients Attending the Snakebite Treatment and Research Hospital in Kaltungo, Northeastern Nigeria

**DOI:** 10.3390/tropicalmed10040103

**Published:** 2025-04-11

**Authors:** Nicholas Amani Hamman, Aashna Uppal, Nuhu Mohammed, Abubakar Saidu Ballah, Danimoh Mustapha Abdulsalam, Frank Mela Dangabar, Nuhu Barde, Bello Abdulkadir, Suraj Abdullahi Abdulkarim, Habu Dahiru, Idris Mohammed, Trudie Lang, Joshua Abubakar Difa

**Affiliations:** 1Snakebite Treatment and Research Hospital, Kaltungo 770110, Gombe State, Nigeria; nuhu2k2@gmail.com (N.M.); abuabus@yahoo.com (A.S.B.); pharmfmdangabar@gmail.com (F.M.D.); bardenuhubarde@gmail.com (N.B.); 2The Global Health Network, Centre for Global Health and Tropical Medicine, Nuffield Department of Medicine, University of Oxford, Oxford OX2 7SG, UK; trudie.lang@ndm.ox.ac.uk; 3Department of Community Medicine and Public Health, Gombe State University, Gombe 760214, Gombe State, Nigeria; mustafabdul2011@gmail.com (D.M.A.); joshuaad50@gsu.edu.ng (J.A.D.); 4Gombe State Hospital Services Management Board, Gombe 760253, Gombe State, Nigeria; abdulkadirbello093@gmail.com; 5State Ministry of Health, Gombe 760253, Gombe State, Nigeria; drsurajkwami@gmail.com (S.A.A.); hodahiru@yahoo.com (H.D.); 6Immunology and Infectious Disease Unit, Federal Teaching Hospital Gombe, Gombe 620261, Gombe State, Nigeria; idris.muhammad42@gmail.com

**Keywords:** snakebite, epidemiology, machine learning, low-resource setting, antivenom, patient outcomes, Nigeria

## Abstract

The Snakebite Treatment and Research Hospital (SBTRH) is a leading centre for snakebite envenoming care and research in sub-Saharan Africa, treating over 2500 snakebite patients annually. Despite routine data collection, routine analyses are seldom conducted to identify trends or guide clinical practices. This study retrospectively analyzes 1022 snakebite cases at SBTRH from January to June 2024. Most patients were adults (62%) and were predominantly male (72%). Key factors such as age, sex, and time between bite and hospital presentation were associated with outcomes, including recovery, amputation, debridement, and death. Adult males who took more than four hours to arrive to hospital were identified as a high-risk group for poor outcomes. Using patient characteristics, an XGBoost model was developed and was compared to Random Forest and logistic regression models. In general, all models had high positive predictive value and low sensitivity, meaning that if they predicted a patient to experience amputation, debridement, or death, that patient almost always actually experienced amputation, debridement, or death; however, most models rarely made this prediction. The XGBoost model with all features was optimal, given that it had both a high positive predictive value and relatively high sensitivity. This may be of significance to resource-limited settings like SBTRH, where antivenoms can be scarce; however, more research is needed to build better predictive models. These findings underscore the need for targeted interventions for high-risk groups, and further research and integration of machine-learning-driven decision support tools in low-resource-limited clinical settings.

## 1. Introduction

Snakebite envenoming, a neglected tropical disease, has emerged as a serious public health crisis in Nigeria and sub-Saharan Africa [1]. Despite its devastating impact, including high morbidity and mortality rates among vulnerable populations and thousands of cases reported annually, it remains largely overlooked [1,2]. Globally, the situation reflects similar neglect, with the World Health Organization (WHO) estimating 81,000–138,000 deaths annually due to snake envenoming—a figure that is likely underestimated due to significant underreporting [3].

Historically, epidemiological investigation into snakebites, particularly those caused by *Echis* spp. (saw-scaled or carpet viper) in the Nigerian savanna region, revealed that bite incidence peaked during the rainy season, which coincided with peak farming activities [4]. Comprehensive data collected from hospitals in Wukari, Bambur, Zaria, Kaltungo, and Gombe, highlight both the long-term and seasonal patterns of these incidents and their impacts on local populations [4,5]. It was not until the 1980s that the true burden of snakebite envenoming became evident, with reports documenting high morbidity and mortality rates in Northern Nigeria [6,7]. The 1990s saw a surge in research efforts, with notable studies [8] highlighting the importance of snakebite envenoming as a significant public health problem. Building on these findings, Warrel et al. [9] emphasized the urgent need for improved data collection, robust research methodologies, and evidence-based policy decisions in combatting this issue.

Recently, there has been a significant shift towards adopting more rigorous and comprehensive approaches to studying snakebite epidemiology in Nigeria. These approaches include advanced methodologies such as systematic reviews, meta-analyses, and mathematical modelling studies. Snakebite epidemiology in Nigeria is marked by a high incidence and mortality rate, with rural communities being disproportionately affected [10,11]. Contributing factors include the country’s geographic location, climate, settlements patterns, and socioeconomic conditions [12].

The Snakebite Treatment and Research Hospital (SBTRH) in Kaltungo, originally known as the Snakebite Ward, has been a key centre for the treatment and research on snakebite envenoming in sub-Saharan Africa for over five decades [5,13]. Handling an average of 2500 cases annually, it is recognized as the largest snakebite hospital in sub-Saharan Africa, in terms of patient volume. Having been in existence since the 1970s, the hospital has played a pivotal role in reducing the burden of snakebite envenoming in the region, achieving high recovery rates [14]. The hospital’s innovative approach, which integrates clinical care, research, and community engagement, aligns closely with the WHO’s recommendation for comprehensive snakebite management [3].

Despite the high recovery rates observed at SBTRH, there is a significant lack of research addressing the critical factors that predict patient recovery in this region. Identifying these factors is essential for optimizing medical practices and interventions related to snakebite envenoming management. Furthermore, the use of machine learning methods for predictive analytics in snakebite research remains underexplored. To bridge this gap, we piloted machine learning approaches to assess their potential in predicting snakebite envenoming outcomes. SBTRH provided an ideal setting for such analyses, offering a vast and high-quality dataset. We aimed to use these data to uncover patient characteristics contributing to the institution’s limited number of patients experiencing complications or poor outcomes.

Accordingly, our objective was to investigate patient characteristics as potential predictors of envenoming outcomes—specifically recovery, amputation, debridement, or death—among snakebite patients treated at SBTRH. To achieve this, we had three distinct aims: (1) to identify key differences in characteristics between patients experiencing recovery versus those experiencing amputation, debridement, or death; (2) to examine the strength of the associations between patient characteristics and envenoming outcomes through logistic regression; and (3) to build an XGBoost classification model to predict envenoming outcomes based on patient characteristics. This comprehensive approach was designed to generate findings that could significantly influence clinical practice and decision making in the treatment of snakebite envenomation in SBTRH.

## 2. Materials and Methods

### 2.1. Data Collection

SBTRH utilizes a paper-based records system, with patient information documented in registers and folders provided by the Gombe State Hospital Services Management Board. For this single-centre retrospective study, data from patient records between 1 January 2024 and 30 June 2024 were entered into a Microsoft Excel 2024 Version 16.83 (Seattle, WA, USA) document. This time-period covered the dry season (from January to March) and the rainy season (from April to June). This six-month period was chosen as it was a time when antivenom was freely and consistently available; from July onwards, antivenom supply was inconsistent, which impacted the number of patients presenting to SBTRH (as they would receive treatment elsewhere) and impacted clinical outcomes among those who presented to SBTRH. Ethical approval was obtained for this study as part of a larger data science capacity building project granted by the Gombe State Ministry of Health.

Patient records provided the following information: the month of snakebite occurrence, patient age, sex, state of origin, occupation, anatomical site of the snakebite, species of snake, number of antivenom vials used, time (in hours) between the snakebite and hospital presentation, and patient outcome at the end of initial admission (death, amputation, debridement, or recovery). The digitized data were cleaned and analyzed using R software 2023 Version 4.3.1. One variable, snake species, had missing data, as not every patient was able to identify the snake responsible for their bite.

### 2.2. Data Analysis

First, sociodemographic and clinical characteristics were summarized for all patients, as well as by treatment outcome. For continuous variables, a Kruskal–Wallis test was used, for categorical variables with cell counts greater or equal to five, a Chi-Squared test was used, and for categorical variables with cell counts less than five, Fisher’s Exact test was used to evaluate differences in patient characteristics between treatment outcomes. Differences were considered to be statistically significant if the corresponding test’s *p*-value was less than 0.05, as per statistical convention. 

Second, both univariate and multivariable logistic regression was used to explore associations between patient characteristics and patient outcomes. We grouped patients who experienced amputation, debridement, or death into one category, due to low numbers of debridements and deaths, and compared them to patients who recovered. Furthermore, we included broad age groups in this model: those aged from 0 to 11 (infant, toddler, childhood), aged from 12 to 17 (adolescent), and aged 18 and over (adult), as per international reporting standards [15]. As well, we converted “number of hours between snakebite and presentation to hospital” into a categorical variable. For the latter, we used the median number of hours between snakebite and presentation to hospital to categorize this variable into two groups: less than the median, and greater than or equal the median. Finally, we considered a sub-analysis where we removed the variable for antivenom dose from both univariate and multivariable logistic regression models. This sub-analysis was undertaken because the causal pathway was not clear; a patient presenting with more severe symptoms, which impacts likelihood of recovery, may be given a high dose of antivenom. This may result in unexpected findings, such as antivenom appearing to promote poor outcomes rather than protect a patient from poor outcomes. We present the results of this sub-analysis in the main text, with additional results presented in the Appendix A, where the antivenom variable was included in both regression models.

Third, we used a machine learning approach to evaluate how well patient characteristics were able to predict patient outcome. A positive outcome, or “case”, was amputation, debridement, or death, whereas a negative outcome, or “control”, was recovery. The same modified dataset that was used for the logistic regression analyses above was used for this as well. Accordingly, we have also provided additional results in the Appendix A concerning the inclusion of the antivenom variable in machine learning models. The XGBoost method was chosen as a binary classification algorithm due to its ease of use, ability to deal with unbalanced classes, and ability to handle clinical data [16,17,18]. At its core, the XGBoost algorithm minimizes the error between the observed and predicted data (through the loss function) and favours reduced model complexity, vis-a-vis regularization [19].

We randomly divided patients in our dataset: 70% of the patient records were used for training the XGBoost model, and 30% were used to test its performance. Classes with less than ten observations were removed from the dataset to ensure reasonable distribution of classes between the test and training dataset. We fixed the hyperparameter “scale_pos_weight”, as our outcome classes were not equal in size; far more patients recovered than those who experienced amputation, debridement, or death. As per convention, we set scale_pos_weight to be equal to the number of patients experiencing amputation, debridement or death, divided by the number of patients who recovered. We also fixed the learning rate at 0.01 to remain conservative. Few other hyperparameters were tuned with Bayesian optimization, with cross-validation over five folds [20]. Table 1 describes the hyperparameters that were tuned and the corresponding ranges of values that were explored through Bayesian optimization. These ranges were informed by hyperparameter values in similar studies [16,17,18,19]. 

Optimized hyperparameters were then used to build an XGBoost model. All features (patient characteristics) were used in the model, as there were only eight features in total; however, a subset of features was subsequently selected based on statistical significance during the previous univariate logistic regression step. This simplified model was constructed using the same Bayesian optimization approach described above. Both models’ performances were evaluated based on the area under the receiver-operator curve (AUROC) statistic. A high AUROC indicated better model performance as this optimized sensitivity and specificity in predicting patient outcome.

To further evaluate the XGBoost models’ performances, their evaluation metrics were compared to those of a class-weighted Random Forest algorithm and a logistic regression algorithm. Hyperparameters for both Random Forest (mtry and ntrees) and logistic regression (mixture and penalty) were also tuned using a Bayesian optimization approach, with ranges informed by the literature, as noted in Table 1. Cross-validation over five folds was used for logistic regression. However, for Random Forest, cross-validation was not used as this model does not need it as a guard for overfitting (the penalty hyperparameter is used instead), but we sampled without replacement to account for unbalanced classes.

## 3. Results

As shown in Table 2, a total of 1022 snakebite cases were seen in the hospital between 1 January 2024 and 30 June 2024. The median age of patients was 20 (interquartile range (IQR) 13–34). Among these, nearly two-thirds were 18 years of age or older (n = 638, 62%), and a majority were male (n = 734, 72%). Most snake bites occurred in April (n = 282, 28%), as this was the beginning of the rainy season, coinciding with commencement of intense farming activity [7,21]. Roughly half of the patients were from Gombe State, which is where SBTRH is located (n = 498, 49%), while the other half of the patients came from neighbouring states, or Cameroon (grouped in with the “Other” category). Nearly half of the patients were farmers (n = 469, 46%). On average, patients presented to hospital within four hours of being bitten (IQR 2–8 h). Most patients were bitten on one of their hands (n = 795, 78%), and most patients were bitten by carpet vipers (n = 810, 79%). Most patients received a single dose of antivenom (n = 664, 65%), while a minority (n = 138, n = 14%) received none due to a clotting time of less than 20 min. Finally, most patients recovered from their snakebite (n = 931, 91%). Of those that did not recover (n = 91, 9%), nine died, seventy-eight experienced amputations, and four underwent debridement surgeries. 

### 3.1. Differences in Patient Characteristics by Treatment Outcome

Also illustrated in Table 2, there were statistically significant differences in patient characteristics by treatment outcome. Proportionally, there were more male patients among those who experienced amputation, debridement, or death (11%) compared to female patients (5%). Further, a proportionally higher number of adults experienced poor outcomes compared to other age groups (11% vs. 8% or 4%). Those who experienced amputation, debridement, or death had a longer time taken to arrive to hospital (median 5 h) compared to those who recovered (median 4 h). Finally, a proportionally higher number of patients who received two or more doses of antivenom experienced amputation, debridement, or death (18%), compared to those who received a single dose (7%) or no antivenom (3%). 

### 3.2. Logistic Regression Analysis

Logistic regression analysis revealed key associations between patients’ characteristics and clinical outcomes (Table 3). According to univariate logistic regression, patients who arrived at SBTRH less than four hours after being bitten, as well as female patients and pediatric patients, were less likely to experience amputation, debridement, or death, compared to their counterparts. However, neither of these three patient characteristics remained statistically significantly associated with poor outcomes in the multivariable regression model. A higher antivenom dose was significantly associated with higher likelihood of experiencing poor outcomes in additional analyses (Appendix A).

### 3.3. XGBoost Model

Finally, the patient data were divided into a training dataset (containing 691 patients), and a test dataset (containing 296 patients). The training dataset was used to build an XGBoost model. The scale_pos_weight hyperparameter was set at 9.85 to reflect the ratio of recoveries to amputations or debridement surgeries. Table 4 describes the variables that were used in both the full and simplified XGBoost models, as well as the hyper parameter values following Bayesian optimization. Appendix A depicts the relationship between the three variables used in the simplified models.

Similarly, a Random Forest model and a logistic regression model were constructed with the full set of features and the simplified set of features. Table 4 lists the hyperparameter values obtained using Bayesian optimization. The results of all models (on the test data) are shown in Table 5.

According to AUROC, which optimizes both sensitivity and specificity, all models had modest performance, regardless of number of features included, with the Random Forest model with all features having the best performance. There were also two models that stood out in terms of sensitivity and specificity: the XGBoost model with the full set of features had high sensitivity and low specificity, and the opposite was true for the Random Forest model with a simplified set of features. In other words, the XGBoost model with the full set of features was better at identifying which patients would experience amputation, debridement, or death and worse at identifying those who would recover. The opposite was true for the Random Forest model with a simplified set of features.

In general, all models had high positive predictive value and low negative predictive value. This means that nearly all patients who were predicted to experience amputation, debridement, or death actually experienced these. However, given low negative predictive values and sensitivity values, there were several patients who actually experienced amputation, debridement, or death that were missed in the models’ predictions. Results were qualitatively similar when the antivenom dose variable was included, except in the case of logistic regression with the full set of features, which did not converge (Appendix A).

## 4. Discussion

This study reports on 1022 cases of snakebite treated at SBTRH Kaltungo from January to June 2024, which we suggest is a representative number that provides insight into the epidemiology, treatment, and outcomes of snakebite incidents seen in the facility, which sees over 2500 cases per year. The data reveal a critical pattern of patient demographics, clinical outcomes, and treatment modalities.

Our findings note that a large proportion of the patients were adults (62%) and were predominantly male (72%), which is consistent with the existing literature [22,23,24]. Further, snakebite incidence peaked in April, at the beginning of the rainy season in this region, when farmers typically clear their farmlands; this is consistent with global and local studies [25,26,27]. Gombe State, where SBTRH is located, accounts for 49% of cases, which is consistent with recent studies showing that Gombe State contributes a higher percentage of cases in the hospital than other states [28,29]. These other studies also noted a predominance of snakebites among male patients (72%) and adults (58%), with farmers (46%) being the most common occupation group. Such trends align with the existing literature [30,31,32,33], which suggests that occupational hazards contribute to snakebite incidents, particularly in agricultural settings. In addition, the male to female ratio reflects societal norms where men are more likely to engage in outdoor labour, thereby increasing their exposure to snakebites. This highlights the need for targeted public health interventions aimed at adult males, particularly in rural areas where agricultural activities predispose individuals to encounters with snakes. For example, interventions may be in the form of educational programmes to increase awareness about snakebite prevention during peak season.

Our study also notes that a majority of the bites occurred on the hand, with 46% on the right hand and 36% on the left hand, which reflects occupational hazards where interactions with snakes occur during manual labour—particularly farming. The site of the bite is crucial in management, as bites from the hands and feet are associated with severe local tissue damage due to its proximity to muscles, tendons, and bones, which predisposes such patients to complications such as necrosis or secondary infection, leading to debridement or amputation.

This study found that 79% of cases were attributed to carpet vipers (Echis romani), which is consistent with previous studies [7]. This is a venomous snake known for its potent hemotoxic venom, which causes severe coagulopathy and local tissue destruction [34]. Notably, 18% of our study’s patients were unable to identify the species of snake that bit them; this is common in snakebite reports, as relatives or patients may not have been able to identify the snake, or the snake may not have been killed and brought to the facility for identification. However, understanding the snake’s species is critical for appropriate and timely administration of antivenom, especially in areas where polyvalent antivenoms are not readily available. Developing a targeted community awareness campaign about risk to farmers and the importance of venomous snake identification would therefore be a beneficial intervention to curb the snakebite burden in this region, and this is a recommendation from our work.

We also note that a significant majority of the patients (91%) recovered, while 9% of cases either ended in amputation, debridement, or death. Despite the high rate of recovery, we identified a trend that raised concern about the severity of some bites, particularly among males, adult patients, and patients who took more than four hours to arrive to hospital, all of whom were more likely to experience complications than their counterparts. Interestingly, these characteristics did not remain significantly associated with patient outcome in a multivariable model, but the association with hours between bite and hospitalization was close to statistical significance and may still be of clinical significance.

The higher proportion of male patients undergoing amputation or debridement or experiencing death may also reflect the occupational risks associated with farming, which is the most common occupation among male patients in this region [1,7]. This emphasizes the need for targeted interventions for these population groups, such as educational interventions to prevent snakebites and emphasize the importance of seeking clinical care. Indeed, shorter time intervals between bite and hospital presentation, as well as belonging to the pediatric age group, were associated with better outcomes in univariate analysis. However, these trends did not remain significant in the multivariable model. This may suggest that other factors—including those that we did not extract from patient records—could have a stronger influence on patient outcomes. Current evidence exploring the associations between age and snakebite severity is inconsistent and limited [35]; in our own dataset, 21% of amputations (n = 78), 50% of debridement surgeries (n = 4), and 55% of deaths (n = 9) were among pediatric patients. Nevertheless, the protective effect of early presentation aligns with the existing literature [29,36], emphasizing the need for timely intervention in snakebite incidents. While patients in our dataset had differences in time taken to present to hospital, all of the patients with a clotting time of 20 min or above received antivenom within one hour of presentation due to free and consistent antivenom during this period; early reception of antivenom is known to improve patient outcomes [37,38].

The differences in rates of recovery between male and female patients and adult and pediatric patients may also reflect in the physiological responses to envenomation, health-seeking behaviour, or access to care. Additionally, a larger proportion of pediatric and male patients received a single dose of antivenom compared to adults and female patients. This raises an important question of whether clinical decision making regarding antivenom administration is influenced by age and sex, or whether these differences reflect underlying severity of envenomation. Further studies could investigate these patterns to underpin specific thresholds for antivenom dosing in relation to envenomation severity; this would ensure equitable and effective care across patient subgroups and different clinical stages of envenomation.

There is a dearth of research that leverages machine learning methods for snakebite epidemiology, although these methods have been previously used for snake identification and geospatial modelling of snake habitats [39,40,41,42]. We sought to apply various machine learning approaches to our institutional dataset, as a proof-of-concept, noting that our dataset was limited in size and breadth of features. Given the data we had, we were able to achieve AUROC values around 0.5–0.6, depending on the machine learning model used. The modest levels of model performance may be due to the relatively low proportion of deaths, amputations, and debridement surgeries in our dataset. This led to a class imbalance in the data, affecting the sensitivity, specificity, positive predictive value, and negative predictive value due to conservative predictions of amputation, debridement, and death. A high positive predictive value coupled with low or modest sensitivity meant that, if a model predicted a patient to experience amputation, debridement, or death, that patient almost always actually experienced amputation, debridement, or death (high positive predictive value); however, the models rarely made this prediction (low sensitivity). Therefore, the XGBoost model with all features was optimal, given that it had both a high positive predictive value and relatively high sensitivity. A high positive predictive value is generally desirable in scenarios where treatment is costly, which, in SBTRH, it often is, due to inconsistent availability of antivenom. However, given that these models often fail to detect high-risk patients, better models would need to be developed to maximize negative predictive values to ensure that high-risk patients are correctly detected and treated in a timely manner. 

While the models developed in this study are not yet useful for screening, they exist as a proof-of-concept that such methodology can be used in these settings, in a similar manner to existing risk-prediction tools [43]. This has potential to be of enormous clinical implication because streamlined models can be implemented in a resource-limited setting like SBTRH, Kaltungo, where rapid, less resource-demanding, and relevant decision making is crucial to reduce snakebite morbidity and mortality. However, more studies of this nature are needed to build more robust machine learning models, as these can be a valuable tool in predicting patient outcomes. Identifying high-risk patients early allows for providers to adjust strategies, thereby improving recovery rates and reducing the need for invasive procedures such as debridement or amputation, or even mortality from snakebites. These findings may support further research leading to the development of risk stratification tools that can guide clinical management of snakebite patients, thereby ensuring effective and efficient use of scarce resources and interventions like antivenoms.

The use of machine learning techniques to analyze large and clinically relevant datasets of snakebites and the application of Bayesian optimization in tuning hyperparameters adds to the robustness of our findings and provides a valuable framework for future studies. Furthermore, the retrospective nature of the study design enabled us to leverage existing institutional data to explore the potential relationships between patient characteristics and clinical outcomes, offering a deeper understanding of the factors influencing snakebite prognosis at our institution. 

However, our study was limited to single centre, which may affect the generalizability of the findings to other regions where snake species, patient demographics, healthcare infrastructure, and type of antivenom may differ. We were also limited by the breadth of the existing data; for example, we were not able to include information on severity of swelling and presence/absence of clotting disorders, which are important factors that can influence patient outcomes. Furthermore, some variables that were routinely collected and complete could have been subject to recall bias (time taken to present to hospital, for example). The limited depth and breadth of the data impacted the utility of machine learning models as well. To enhance the external validity of the findings of our studies, future studies should look at multicentre collaboration using a standardized protocol across different healthcare settings. 

Future studies should aim at validating our findings and incorporating additional clinical and laboratory data in a multicentre study to enhance predictability and generalizability of the predictive models. Efforts should also be made at implementing machine-learning-driven decision-support tools in real-time clinical settings to improve patient outcomes.

## 5. Conclusions

Conclusively, our study is able to identify several key predictors of clinical outcomes such as sex, age group, and time before presentation at the hospital. It also demonstrates that further research is needed to build better machine learning models such as XGBoost, Random Forest, and logistic regression, which could be used to enhance clinical decision making in resource limited settings.

The findings of this study have significant implications for public health practice. These include the development of targeted interventions for high-risk groups (i.e., males, farmers, and individuals living in rural areas), optimization of antivenom administration for high-risk patients (adult males who have a long travel time to reach the hospital), and potential integration of machine learning-driven decision-support tools into clinical settings. Furthermore, policymakers should prioritize resource allocation for snakebite prevention and treatment programmes, particularly targeting high-risk groups. Further validation of these findings is needed in a multicentre study to support the integration of predictive models into routine clinical practice, especially in resource-limited settings, which usually have the highest snakebite incidence in the world.

A key finding that can be taken up into an immediate output is an education campaign with two elements. Firstly, to communicate the risk and risk reduction mechanisms for farmers, and secondly, guidance and education on the importance of snake identification. We are going to share our findings in regional channels and also share with them the global snakebite research community through The Global Health Network [44]. Our study also raises important remaining unknowns, namely, that we need more and better data in a multicentre study to improve on machine learning model performance for predicting patient outcomes.

We undertook this study within a larger research methodology study [45] that is ongoing and seeks to find effective capacity development mechanisms to support care settings, such as SBTRH, to analyze the data they hold to guide their management and treatment practices. Therefore, we shall endeavour to run further studies to address the remaining questions and encourage others to do the same.

## Figures and Tables

**Table 1 tropicalmed-10-00103-t001:** Hyperparameters tuned for machine learning algorithms.

Hyperparameter	Range of Values
XGBoost	
subsample: the proportion of all patients sampled for each tree	[0.25, 1]
max_depth: the maximum depth of each tree (integer value)	[2, 10]
min_child: the lowest sum of weights of a child node	[1, 25]
Random Forest	
mtry: the number of predictor variables selected at each tree split	[1, 3]
ntrees: the number of decision trees to be built	[10, 40]
Logistic Regression	
mixture: controls balance between lasso and ridge regularization	[0, 1]
penalty: controls how much shrinkage is applied to model coefficients	[0, 1]

**Table 2 tropicalmed-10-00103-t002:** Patient characteristics by outcome.

Variable	Overall, N = 1022 ^1^	Treatment Outcome		*p*-Value ^2^
		Amputation, Debridement or Death, N = 91 ^1^	Recovery, N = 931 ^1^	
Month of Snakebite Occurrence				0.5
April	282	27 (10%)	255 (90%)	
March	213	23 (11%)	190 (89%)	
June	183	14 (8%)	169 (92%)	
May	180	10 (6%)	170 (94%)	
February	95	10 (11%)	85 (89%)	
January	69	7 (10%)	62 (90%)	
Age	20	22 (18, 35)	20 (13, 34)	0.082
Age Group				0.020
Adult (18+)	638	68 (11%)	570 (89%)	
Adolescent (12 to 17)	196	15 (8%)	181 (92%)	
Infant, toddler, childhood (0 to 11)	188	8 (4%)	180 (96%)	
Sex				0.002
Male	734	78 (11%)	656 (89%)	
Female	288	13 (5%)	275 (95%)	
State or Country of Origin				
Gombe	498	34 (7%)	464 (93%)	
Taraba	188	13 (7%)	175 (93%)	
Adamawa	143	16 (11%)	127 (89%)	
Bauchi	98	17 (17%)	81 (83%)	
Borno	59	7 (12%)	52 (88%)	
Yobe	30	4 (13%)	26 (87%)	
Other ^3^	6	0 (0%)	6 (100%)	
Occupation				
Farmer	469	59 (13%)	410 (87%)	
Under Care	215	12 (6%)	203 (94%)	
House Wife	152	8 (5%)	144 (95%)	
Student	138	9 (7%)	129 (93%)	
Business	36	3 (8%)	33 (92%)	
Civil Servant	8	0 (0%)	8 (100%)	
Other ^3^	4	0 (0%)	4 (100%)	
Hours Between Bite and Hospitalization	4 (2, 8)	5 (4, 8)	4 (2, 8)	0.014
Site of Snakebite				>0.9
Upper Limb	795	72 (9%)	723 (91%)	
Lower Limb	224	19 (8%)	205 (92%)	
Other ^3^	3	0 (0%)	3 (100%)	
Snake Species				>0.9
Carpet Viper (Echis romani)	810	74 (9%)	736 (91%)	
Unidentifiable	188	17 (9%)	171 (91%)	
Cobra (Naja)	10	0 (0%)	10 (100%)	
Night Adder (Causus rhombeatus)	7	0 (0%)	7 (100%)	
Mole Viper (Atractaspididae)	5	0 (0%)	5 (100%)	
Other ^3^	2	0 (0%)	2 (100%)	
Antivenom Dose (Number of Vials)				<0.001
1	664	48 (7%)	616 (93%)	
2 or more	220	39 (18%)	181 (82%)	
0	138	4 (3%)	134 (97%)	

^1^ Median (Q1, Q2) or frequency (%). ^2^ Due to small numbers/zeroes in some categories, for occupation and state of origin, a *p*-value was not calculated. ^3^ Categories with less than five individuals across age groups have been grouped into ‘Other’ to protect patient privacy.

**Table 3 tropicalmed-10-00103-t003:** Logistic regression for outcome (likelihood of experiencing amputation, debridement, or death, compared to recovery).

Characteristic	Univariate Models		Multivariate Model	
	OR ^1^	95% CI ^1^	Adjusted OR ^1^	95% CI ^1^
Month of Snakebite Occurrence				
January	—	—	—	—
February	1.04	0.38, 3.01	1.07	0.38, 3.19
March	1.07	0.46, 2.81	0.89	0.36, 2.42
April	0.94	0.41, 2.43	0.87	0.36, 2.31
May	0.52	0.19, 1.49	0.41	0.15, 1.22
June	0.73	0.29, 2.01	0.58	0.22, 1.64
Age Group				
Infant, toddler, childhood (0 to 11)	—	—	—	—
Adolescent (12 to 17)	1.86	0.79, 4.73	2.09	0.71, 6.29
Adult (18+)	2.68	1.34, 6.15	3.22	0.97, 11.7
Sex				
Female	—	—	—	—
Male	2.52	1.42, 4.81	1.82	0.81, 4.56
State or Country of Origin				
Adamawa	—	—	—	—
Bauchi	1.67	0.79, 3.51	1.82	0.84, 3.95
Borno	1.07	0.39, 2.66	1.04	0.37, 2.65
Gombe	0.58	0.32, 1.11	0.82	0.39, 1.75
Other	0.00	0.00, Inf	0.00	0.00, Inf
Taraba	0.59	0.27, 1.27	0.60	0.27, 1.32
Yobe	1.22	0.33, 3.66	1.06	0.28, 3.27
Occupation				
Business	—	—	—	—
Civil Servant	0.00	0.00, Inf	0.00	0.00, 0.00
Farmer	1.58	0.55, 6.72	1.53	0.50, 6.65
House Wife	0.61	0.17, 2.90	0.92	0.20, 5.26
Other	0.00	0.00, Inf	0.00	0.00, Inf
Student	0.77	0.21, 3.60	1.20	0.31, 6.03
Under Care	0.65	0.19, 2.96	1.58	0.32, 9.41
Site of Snakebite				
Lower Limb	—	—	—	—
Other	0.00	0.00, Inf	0.00	0.00, Inf
Upper Limb	1.07	0.65, 1.87	1.07	0.63, 1.91
Snake Species				
Carpet Viper (Echis romani)	—	—	—	—
Cobra (Naja)	0.00	0.00, Inf	0.00	0.00, Inf
Mole Viper (Atractaspididae)	0.00	0.00, Inf	0.00	0.00, Inf
Night Adder (Causus rhombeatus)	0.00	0.00, Inf	0.00	0.00, Inf
Other	0.00	0.00, Inf	0.00	0.00, Inf
Unidentifiable	0.99	0.55, 1.68	1.01	0.55, 1.76
Hours Between Bite and Hospitalization				
4 h or more	—	—	—	—
Less than 4 h	0.50	0.30, 0.80	0.57	0.29, 1.11

^1^ OR = odds ratio, CI = confidence interval.

**Table 4 tropicalmed-10-00103-t004:** Machine learning model features and hyperparameter values.

	Full Model	Simplified Model
Features	Month of snakebite occurrence State or country of origin Sex Occupation Site of snakebite Snake species Age category (pediatric vs. adult) Hours between bite and hospitalization (above vs. below median value)	Sex Age category (pediatric vs. adult) Hours between bite and hospitalization (above vs. below median value)
Hyperparameter values	XGBoost	
	max_depth = 9 min_child_weight = 5.86 subsample = 0.26	max_depth = 6 min_child_weight = 24.49 subsample = 0.86
	Random Forest	
	mtry = 2 ntrees = 10	mtry = 3 ntrees = 10
	Logistic Regression	
	mixture = 0 penalty = 0	mixture = 0 penalty = 0

**Table 5 tropicalmed-10-00103-t005:** Machine learning model results.

Features	Model	Sensitivity	Specificity	Positive Predictive Value	Negative Predictive Value	AUROC
Full Set	XGBoost	0.80	0.26	0.94	0.08	0.532
	Random Forest	0.58	0.58	0.95	0.09	0.582
	Logistic Regression	0.58	0.42	0.94	0.07	0.505
Simplified (Three Features)	XGBoost	0.53	0.53	0.94	0.07	0.529
	Random Forest	0.08	0.95	0.95	0.07	0.512
	Logistic Regression	0.53	0.53	0.94	0.07	0.529

## Data Availability

Data are available upon reasonable request to the corresponding authors.

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
