# Peer review of "Using a Machine Learning Approach to Predict Snakebite Envenoming Outcomes Among Patients Attending the Snakebite Treatment and Research Hospital in Kaltungo, Northeastern Nigeria"

_tropicalmed, 2025, doi:10.3390/tropicalmed10040103_

Round 1
Reviewer 1 Report
Comments and Suggestions for Authors
Overall Assessment
This is a very interesting article describing and modelling snakebite outcomes in a hospital in Northern Nigeria. The article is well written, and the methods are generally appropriate. However, I have a few comments, particularly regarding the use and interpretation of model performance indicators, which I hope the authors can address.
Specific comments:
- Line 202–203: The authors report that a higher proportion of adults experienced poor outcomes compared to pediatric patients. Could the authors comment on the possibility of survival bias? Specifically, could it be that severely affected children are more likely to die before reaching the hospital? This could be the case if venom acts more rapidly in children than in adults.
- Antivenom Administration: The authors state that 14% of patients did not receive antivenom. Could they clarify the reasons for this? Was it due to a shortage of antivenom, clinical decision-making, or other factors?
- Seasonality of Snakebite Incidence:
- Previous studies cited on line 47 indicate that snakebite incidence peaks during the rainy season (June–September according to WorldBank website).
- This study, which includes data from January to June, reports a peak in April (followed by March and June).
- On line 265, the authors state that this peak coincides with the start of the rainy season.
- Clarification needed: Does snakebite incidence peak at the beginning of the rainy season or during it? If it peaks during the rainy season, then the current dataset (January–June) may not be sufficient to conclude on seasonality.
- Tables 2 and 3: Consider combining these tables in R using gt::add_overall() to improve readability.
- Definition of the Positive Outcome: The methods section should explicitly state that the positive outcome of interest is "amputation, debridement or death (ADD).” Currently, this information is only found in the caption of Table 4, but it is essential for correctly interpreting sensitivity (SE), specificity (SP), positive predictive value (PPV), and negative predictive value (NPV).
- Table 6 – Model Performance Values: It is interesting to note that PPV is consistently high and NPV low, while SE/SP varies in a balanced way from one model to another (the higher the SE, the lower the SP, and vice versa).
- Lines 252–255 : The sentence “(...) it achieved perfect specificity, meaning a patient’s age, sex, and time taken to arrive to hospital could perfectly predict if they were going to experience amputation, debridement, or death. However, this came at a cost of having very low sensitivity, meaning these characteristics were unlikely to accurately predict recovery.” is misleading because it refers to SP and SE instead of PPV and NPV and it suggests that, with this model, the clinician could perfectly well predict whether the patient would undergo amputation, debridement or death (ADD). Actually, the combination of 100% SP and PPV with a very low (<10%) SE and NPV means that:
- The model only predicts ADD when it is absolutely certain, but it misses almost all other ADD cases (low sensitivity).
- Every patient predicted to have ADD actually experienced ADD, but the model rarely makes this prediction (high PPV, low sensitivity).
- Among those predicted to survive, a large proportion actually experienced ADD (low NPV).
- As such, it is important to explain to the reader that this model has very limited utility for screening or triage since it fails to detect most high-risk patients.
- This could indicate overfitting to a small subset of extreme cases or an imbalance in the dataset, where the model learns to be highly conservative in predicting ADD.
- Clinical Relevance of Predictive Values:
- Since the predictive model is intended for screening patients upon admission, PPV and NPV are more relevant than SE and SP. The authors should emphasise predictive values when discussing model performance.
- Furthermore, while the authors acknowledge that the model does not perform well enough for screening, they should specify what level of PPV and NPV would be considered acceptable in this clinical setting.
- For instance:
- A high PPV is desirable when the treatment is costly or has significant risks.
- A moderate PPV (allowing more false positives) is acceptable if the treatment is inexpensive or has minimal harm.
- A high NPV is important if the condition progresses quickly and effective treatment exists if taken early on.
- A moderate NPV might be acceptable if the condition progresses slowly or if early intervention has limited benefits.
- I'm not familiar with snakebite case management, but it seems to me that a more useful screening tool should achieve a high NPV and at least moderate PPV, which is clearly not the case here.
- Abstract (Line 30): The word “robust” is vague in this context. I suggest replacing it with “predictive”, which more accurately describes the model's function.
- Hyperparameter Tuning (Lines 239–243): The authors should provide more details on how the hyperparameters for the random forest and logistic regression models were selected (or optimised), as they did for the XGBoost (line 232).
Anton CAMACHO
Author Response
Overall Assessment
This is a very interesting article describing and modelling snakebite outcomes in a hospital in Northern Nigeria. The article is well written, and the methods are generally appropriate. However, I have a few comments, particularly regarding the use and interpretation of model performance indicators, which I hope the authors can address.
Response: Thank you for your insightful suggestions that will greatly improve the quality of our manuscript!
Specific comments:
- Line 202–203: The authors report that a higher proportion of adults experienced poor outcomes compared to pediatric patients. Could the authors comment on the possibility of survival bias? Specifically, could it be that severely affected children are more likely to die before reaching the hospital? This could be the case if venom acts more rapidly in children than in adults.
Response: Current evidence exploring the relationship between patient age and snakebite envenomation andoutcome especially between Paediatric age group and Adult is both variable and limited. We have added to our discussion on lines 324-327: “Current evidence exploring associations between age and snakebite severity is inconsistent and limited [34]; in our own dataset, 21% of amputations (n = 78), 50% of debridement surgeries (n = 4), and 55% of deaths (n = 9) were among paediatric patients.”
- Antivenom Administration: The authors state that 14% of patients did not receive antivenom. Could they clarify the reasons for this? Was it due to a shortage of antivenom, clinical decision-making, or other factors?
Response: We have added to our results accordingly on lines 200-202: “Most patients received a single dose of antivenom (n = 664, 65%), while a minority (n = 138, n = 14%) received none, due to a clotting time of less than 20 minutes for Carpet viper envenomed patients.”
- Seasonality of Snakebite Incidence: Previous studies cited on line 47 indicate that snakebite incidence peaks during the rainy season (June–September according to WorldBank website). This study, which includes data from January to June, reports a peak in April (followed by March and June). On line 265, the authors state that this peak coincides with the start of the rainy season. Clarification needed: Does snakebite incidence peak at the beginning of the rainy season or during it? If it peaks during the rainy season, then the current dataset (January–June) may not be sufficient to conclude on seasonality.
Response: Snakebite incidence peaks at the onset of the rainy season. We have clarified wording in our methods on lines 101-103: “This time-period covered the dry season (January through March) and the rainy season (April through June).”. We have changed the wording in lines 194-196 accordingly and have added two references: “Most snake bites occurred in April (n = 282, 28%), as this was the beginning of the rainy season, coinciding with commencement of intense farming activity [21,22].”. As well, in our discussion, on lines 273-276, we have noted: “Further, snakebite incidence peaked in April, at the beginning of the rainy season in this region, when farmers typically clear their farmlands; this is consistent with global and local studies [26–28].”.
- Tables 2 and 3: Consider combining these tables in R using gt::add_overall() to improve readability.
Response: We have now combined these tables (see Table 2).
- Definition of the Positive Outcome: The methods section should explicitly state that the positive outcome of interest is "amputation, debridement or death (ADD).” Currently, this information is only found in the caption of Table 4, but it is essential for correctly interpreting sensitivity (SE), specificity (SP), positive predictive value (PPV), and negative predictive value (NPV).
Response: We have added this detail on lines 148-149: “A positive outcome, or “case” was amputation, debridement, or death, whereas a neg-ative outcome, or “control”, was recovery”.
- Table 6 – Model Performance Values: It is interesting to note that PPV is consistently high and NPV low, while SE/SP varies in a balanced way from one model to another (the higher the SE, the lower the SP, and vice versa).
Response: We have modified our results section to include this note, on lines 253-265: “According to AUROC, which optimizes both sensitivity and specificity, all models had modest performance, regardless of number of features included, with the ran-dom-forest model with all features having the best performance. Random-forest models also stood out in terms of sensitivity and specificity; while other models tended to have relatively equal sensitivity than specificity, random-forest models had higher specificity than sensitivity, meaning they were better at predicting which patients would experience amputation, debridement, or death, and worse at predicting who would recover. In general, all models had high positive predictive value and low negative pre-dictive value. This means that nearly all patients who were predicted to experience amputation, debridement, or death, actually experienced these. However, given low negative predictive values and sensitivity values, there were several patients who ac-tually experienced amputation, debridement, or death, that were missed in the models’ predictions. Results were qualitatively similar when the antivenom dose variable was included (Table S2).”
- Lines 252–255 : The sentence “(...) it achieved perfect specificity, meaning a patient’s age, sex, and time taken to arrive to hospital could perfectly predict if they were going to experience amputation, debridement, or death. However, this came at a cost of having very low sensitivity, meaning these characteristics were unlikely to accurately predict recovery.” is misleading because it refers to SP and SE instead of PPV and NPV and it suggests that, with this model, the clinician could perfectly well predict whether the patient would undergo amputation, debridement or death (ADD). Actually, the combination of 100% SP and PPV with a very low (<10%) SE and NPV means that:
- The model only predicts ADD when it is absolutely certain, but it misses almost all other ADD cases (low sensitivity).
- Every patient predicted to have ADD actually experienced ADD, but the model rarely makes this prediction (high PPV, low sensitivity).
- Among those predicted to survive, a large proportion actually experienced ADD (low NPV).
- As such, it is important to explain to the reader that this model has very limited utility for screening or triage since it fails to detect most high-risk patients.
- This could indicate overfitting to a small subset of extreme cases or an imbalance in the dataset, where the model learns to be highly conservative in predicting ADD.
Response: We have removed that sentence and have instead modified our results section as noted in the previous response. In addition, we have modified our discussion section, on lines 350363: “This led to a class imbalance in the data, affecting the sensitivity, specificity, positive predictive value, and negative predictive value, due to conservative predictions of amputation, debridement, and death. A high positive predictive value coupled with low sensitivity meant that if a model predicted a patient to experience amputation, deb-ridement, or death, that patient almost always actually experienced amputation, deb-ridement, or death (high positive predictive value), however, the models rarely made this prediction (low sensitivity). A high positive predictive value is generally desirable in scenarios where treatment is costly, which, in SBTRH, it often is, due to inconsistent availability of antivenom. However, given that these models often fail to detect high-risk patients, better models would need to be developed to maximize negative predictive values to ensure that high-risk patients are correctly detected and treated in a timely manner. While the models developed in this study are not yet useful for screening, they exist as a proof-of-concept that such methodology can be used in these settings, in a similar manner to existing risk-prediction tools [44].” We have also modified our abstract to include some of this wording.
- Clinical Relevance of Predictive Values:
- Since the predictive model is intended for screening patients upon admission, PPV and NPV are more relevant than SE and SP. The authors should emphasise predictive values when discussing model performance.
- Furthermore, while the authors acknowledge that the model does not perform well enough for screening, they should specify what level of PPV and NPV would be considered acceptable in this clinical setting.
- For instance:
- A high PPV is desirable when the treatment is costly or has significant risks.
- A moderate PPV (allowing more false positives) is acceptable if the treatment is inexpensive or has minimal harm.
- A high NPV is important if the condition progresses quickly and effective treatment exists if taken early on.
- A moderate NPV might be acceptable if the condition progresses slowly or if early intervention has limited benefits.
- I'm not familiar with snakebite case management, but it seems to me that a more useful screening tool should achieve a high NPV and at least moderate PPV, which is clearly not the case here.
Response: As above, we have modified our discussion section, and have included on lines 356-358: “A high positive predictive value is generally desirable in scenarios where treatment is costly, which, in SBTRH, it often is, due to inconsistent availability of antivenom.”
- Abstract (Line 30): The word “robust” is vague in this context. I suggest replacing it with “predictive”, which more accurately describes the model's function.
Response: We have modified lines 29-31: “This may be of significance to resource-limited settings like SBTRH, where antivenoms can be scarce, however, more research is needed to build better predictive models.”
- Hyperparameter Tuning (Lines 239–243): The authors should provide more details on how the hyperparameters for the random forest and logistic regression models were selected (or optimised), as they did for the XGBoost (line 232).
Response: We have added more detail to Table 1 and Table 4, as well as lines 183-189: “Hyperparameters for both random-forest (mtry and ntrees) and logistic regression (mixture and penalty) were also tuned using a Bayesian Optimization approach, with ranges informed by literature, as noted in Table 1. Cross validation over five folds was used for logistic regression. However, for random-forest, cross validation was not used as this model does not need it as a guard for overfitting (the penalty hyperparameter is used instead), but we sampled without replacement to account for unbalanced classes.”
Reviewer 2 Report
Comments and Suggestions for Authors
The authors present an analysis of 1.022 cases of snakebites and risk factors for poor outcomes in Gombe State, Nigeria. Additionally, they employ a machine learning approach to develop predictive models to identify individuals at risk for poor outcome. While the analysis and the data it contains are valuable contributions to the existing body of knowledge on snakebite envenoming in Africa, I have some reservations regarding the accuracy and practicality of the predictive model for clinical use. The manuscript would benefit from further clarification on several points before it can be published. Aditionally, a machine learning expert with a stronger background in the field should be consulted to assess the accuracy of the models.
Key points:
Data collection:
The decision to collect data for a mere six months appears unsual, given the highly seasonal nature of snakebites and the potential variation in risk factors (e.g. snake species) over the course of a year. This may lead to an misestimation of outcomes.
Outcome:
The choice of a combined outcome endpoint is understandable, but the difference between death and wound debridement is significant. The rationale for selecting this endpoint is not clear, and the number of patients who died, underwent amputation or had debridement should be specified to facilitate the interpretation of the findings. The significance of amputation is also unclear, as it is not specified whether digits or limbs were amputated. The authors should explain the choice of the endpoint and describe how many patients were in each category. In addition, the authors should discuss what influence the choice of the endpoint may have had on the performance of the predictive models. It is also not clear when this endpoint was measured. Was this only within the initial admission? Could patients have returned for debridement later? Could these patients have been missed?
Risk factors:
Age: It is unclear which age cut-off was chosen and why (only adult vs. pediatric). Generally, children are at a higher risk of poor outcomes due to their lower body weight; however, this is not the case here, and the authors do not present reasons for this discrepancy. The authors should consider whether the cut-off between adults and children may be too high. If the cut-off is too high, this could lead to an underestimation of the risk in children.
The authors should also present the age distribution of the population to facilitate a more accurate interpretation.
Time to admission: The authors have identified the time between bite and admission as a risk factor, but this is often very inaccurate as patients often cannot indicate exactly when they were bitten. Could this also be the case in this data set? Furthermore, the time to treatment may be more important and may not correlate well with time to admission, depending on the setting. Treatment may be provided more quickly during the day than at night, and the time of arrival in the hospital may be important. The authors should consider including this variable in the dataset. Alternatively, they could measure the time to treatment (antivenom) instead of the time to admission. If this is not applicable to the setting, the authors should at least discuss this. It would also be beneficial to explore whether the 4-hour cut-off is the most appropriate. It seems plausible that poor outcomes are more likely to occur with even later presentations. This is, of course, speculative, but showing more detailed distribution of the data prior to the establishment of cut-offs would facilitate more straightforward interpretation.
Additional risk factors:
The presence of clotting disorders at the time of presentation, and the severity of the initial swelling, may be more indicative than the analysed variables. Can these factors be extracted from the data? If not, this should be discussed.
Predictive modelling:
In general, it would be beneficial to have a reviewer with greater experience in machine learning reviewing this in addition to myself. Nevertheless, it is unclear to me how these models should be useful in clinical management. The authors examine only a limited number of personal risk factors and attempt to construct a predictive model, yet the efficacy of these models is ultimately constrained by the quality of the input data. While the motivation to develop models employing machine learning algorithms is evident, the utility of these models appears limited in this context.
The authors could have systematically collected data over a full year retrospectively, thereby effectively doubling the size of the training set.
As the current models are inadequate (as shown by the relatively low AUC and stated by the authors), the authors should consider removing the modelling approach and only present the analysis of the data. At least, the weaknesses and how they can be improved should be added in the discussion.
Minor points:
The snakes in the tables differ. One contains 7 cases of Puff adder bites, one contains 7 cases of Night adder bites.
In the logistic regression model, a single p-value for each variable would suffice, as there is no requirement to indicate a p-value for every single category.
The authors could consider summarising the site of bite to upper and lower limb, as there is no reason why bites to one side of the body should be more severe. However, bites to the peripheries of the extremities may be more likely to cause necrosis that leads to amputation.
The authors' conclusion that "This emphasises the need for targeted and more aggressive interventions for these population groups, such as closer monitoring, early surgical intervention and careful repeated dosing of antivenoms" appears to be somewhat misguided. Is debridement not an early surgical intervention? The implementation of further surgical interventions in this context would have resulted in an increased number of poor outcomes. Furthermore, the repeated administration of antivenom may not have been adequate. If the relevant risk factor is time to treatment, repeating treatments later on is unlikely to lead to improved outcomes, a factor that is especially true for cases of mainly cytotoxic envenoming.
I disagree with the statement in the discussion that “Furthermore, the retrospective nature of the study design enabled us to explore a diverse array of patient characteristics and clinical outcomes, offering a comprehensive understanding of factors influencing snakebite prognosis“ The authors only analysed a small number of variables and do not present a comprehensive understanding of factors influencing prognosis. This should be phrased in a more cautious way.
Author Response
The authors present an analysis of 1.022 cases of snakebites and risk factors for poor outcomes in Gombe State, Nigeria. Additionally, they employ a machine learning approach to develop predictive models to identify individuals at risk for poor outcome. While the analysis and the data it contains are valuable contributions to the existing body of knowledge on snakebite envenoming in Africa, I have some reservations regarding the accuracy and practicality of the predictive model for clinical use. The manuscript would benefit from further clarification on several points before it can be published. Aditionally, a machine learning expert with a stronger background in the field should be consulted to assess the accuracy of the models.
Response: We greatly appreciate the time taken to review our manuscript and the helpful suggestions in improving its quality!
Key points:
Data collection:
The decision to collect data for a mere six months appears unsual, given the highly seasonal nature of snakebites and the potential variation in risk factors (e.g. snake species) over the course of a year. This may lead to an misestimation of outcomes.
Response: We chose this six month period as antivenoms were freely and consistently available during this time. We’ve added some explanation on lines 103-106: “This six-month period was chosen as it was a time when antivenom was freely and consistently available; from the third week of July onwards, there was antivenom scarcity as the free antivenoms got exhausted and there was also scarcity in the open market which significantly served as the confounding factor that impacted clinical outcomes among those who presented to SBTRH.”
Outcome:
The choice of a combined outcome endpoint is understandable, but the difference between death and wound debridement is significant. The rationale for selecting this endpoint is not clear, and the number of patients who died, underwent amputation or had debridement should be specified to facilitate the interpretation of the findings. The significance of amputation is also unclear, as it is not specified whether digits or limbs were amputated. The authors should explain the choice of the endpoint and describe how many patients were in each category. In addition, the authors should discuss what influence the choice of the endpoint may have had on the performance of the predictive models. It is also not clear when this endpoint was measured. Was this only within the initial admission? Could patients have returned for debridement later? Could these patients have been missed?
Response: We have added the following detail to line 113: “...patient outcome at the end of initial admission…”. Further, the rationale for combining deaths with amputations and debridements was that there were very few deaths (and very few debridements for that matter), and analyses that separated these outcomes had nonsensical results. We’ve added details to clarify on lines 129-131: “We grouped patients who experienced amputation, debridement, or death into one category, due to low numbers of debridements and deaths, and compared them to pa-tients who recovered”.
Risk factors:
Age: It is unclear which age cut-off was chosen and why (only adult vs. pediatric). Generally, children are at a higher risk of poor outcomes due to their lower body weight; however, this is not the case here, and the authors do not present reasons for this discrepancy. The authors should consider whether the cut-off between adults and children may be too high. If the cut-off is too high, this could lead to an underestimation of the risk in children.
Response: Current evidence exploring the relationship between patient age and snakebite envenomation severity is both variable and limited. We have added to our discussion on lines 324-327: “Current evidence exploring associations between age and snakebite severity is inconsistent and limited [34]; in our own dataset, 21% of amputations (n = 78), 50% of debridement surgeries (n = 4), and 55% of deaths (n = 9) were among paediatric patients.” We have also added to lines 133-134: “...adults were defined as those 18 years or older, as per international reporting standards [15].”
The authors should also present the age distribution of the population to facilitate a more accurate interpretation.
Response: We have added on lines 192-193: “The median age of patients was 20 (interquartile range (IQR) 13 – 34)” and have added this detail to Table 2.
Time to admission: The authors have identified the time between bite and admission as a risk factor, but this is often very inaccurate as patients often cannot indicate exactly when they were bitten. Could this also be the case in this data set? Furthermore, the time to treatment may be more important and may not correlate well with time to admission, depending on the setting. Treatment may be provided more quickly during the day than at night, and the time of arrival in the hospital may be important. The authors should consider including this variable in the dataset. Alternatively, they could measure the time to treatment (antivenom) instead of the time to admission. If this is not applicable to the setting, the authors should at least discuss this. It would also be beneficial to explore whether the 4-hour cut-off is the most appropriate. It seems plausible that poor outcomes are more likely to occur with even later presentations. This is, of course, speculative, but showing more detailed distribution of the data prior to the establishment of cut-offs would facilitate more straightforward interpretation.
Response: We have added the following detail to lines 329-332: “While patients in our dataset had differences in time taken to present to hospital, all patients with clotting time of 20 minutes or above received antivenom within one hour of presentation due to free and consistent antivenom during this period; early reception of antivenom is known to improve patient outcomes [38,39].”. We have also added lines 384-386: “We were also limited by the breadth of existing data, and some variables that were routinely collected and complete could have been subject to recall bias (time taken to present to hospital, for example).”
Additional risk factors:
The presence of clotting disorders at the time of presentation, and the severity of the initial swelling, may be more indicative than the analysed variables. Can these factors be extracted from the data? If not, this should be discussed.
Response: Unfortunately, presence of clotting disorders is not routinely reported. The 20-minute blood clotting test is the typical diagnostic method, which is routinely done. In addition, the presence or absence of swelling is collected, but not the severity of swelling. Nearly all patients presented with swelling, so this was not a meaningful addition to the dataset.
Predictive modelling:
In general, it would be beneficial to have a reviewer with greater experience in machine learning reviewing this in addition to myself. Nevertheless, it is unclear to me how these models should be useful in clinical management. The authors examine only a limited number of personal risk factors and attempt to construct a predictive model, yet the efficacy of these models is ultimately constrained by the quality of the input data. While the motivation to develop models employing machine learning algorithms is evident, the utility of these models appears limited in this context.
Response: We have revised our results as per Reviewer 1’s suggestions. In general, wording has been loosened. See lines 356-363: “A high positive predictive value is generally desirable in scenarios where treatment is costly, which, in SBTRH, it often is, due to inconsistent availability of antivenom. However, given that these models often fail to detect high-risk patients, better models would need to be developed to maximize negative predictive values to ensure that high-risk patients are correctly detected and treated in a timely manner. While the models developed in this study are not yet useful for screening, they exist as a proof-of-concept that such methodology can be used in these settings, in a similar manner to existing risk-prediction tools [44].”
The authors could have systematically collected data over a full year retrospectively, thereby effectively doubling the size of the training set.
Response: As detailed in a previous response, from July onwards, antivenom supply was inconsistent, which impacted the number of patients presenting to SBTRH (as they would receive treatment elsewhere) and impacted clinical outcomes among those who presented to SBTRH. Accordingly, we did not expand our dataset to include patients from July through December.
As the current models are inadequate (as shown by the relatively low AUC and stated by the authors), the authors should consider removing the modelling approach and only present the analysis of the data. At least, the weaknesses and how they can be improved should be added in the discussion.
Response: We strongly believe that leaving the predictive modelling in this study is important as it serves as a proof-of-concept. We’ve loosened wording and conclusions, and have noted that while model performance is adequate in our study, with more data, these models could be useful in clinical decision making, particularly in resource-limited settings.
Minor points:
The snakes in the tables differ. One contains 7 cases of Puff adder bites, one contains 7 cases of Night adder bites.
Response: This was an error; we’ve corrected all cases to Night Adder.
In the logistic regression model, a single p-value for each variable would suffice, as there is no requirement to indicate a p-value for every single category.
Response: In order to avoid confusion, we’ve removed p-values from this table, as 95% Confidence Intervals are sufficient for interpretation (i.e. if the interval overlaps 1, the odds ratio is not statistically significant).
The authors could consider summarising the site of bite to upper and lower limb, as there is no reason why bites to one side of the body should be more severe. However, bites to the peripheries of the extremities may be more likely to cause necrosis that leads to amputation.
Response: We have made this change in Tables 2, 3, and S1.
The authors' conclusion that "This emphasises the need for targeted and more aggressive interventions for these population groups, such as closer monitoring, early surgical intervention and careful repeated dosing of antivenoms" appears to be somewhat misguided. Is debridement not an early surgical intervention? The implementation of further surgical interventions in this context would have resulted in an increased number of poor outcomes. Furthermore, the repeated administration of antivenom may not have been adequate. If the relevant risk factor is time to treatment, repeating treatments later on is unlikely to lead to improved outcomes, a factor that is especially true for cases of mainly cytotoxic envenoming.
Response: We have changed lines 317-319 to read: “This emphasizes the need for targeted interventions for these population groups, such as educational interventions to prevent snakebites and emphasize the importance of seeking clinical care.”
I disagree with the statement in the discussion that “Furthermore, the retrospective nature of the study design enabled us to explore a diverse array of patient characteristics and clinical outcomes, offering a comprehensive understanding of factors influencing snakebite prognosis“ The authors only analysed a small number of variables and do not present a comprehensive understanding of factors influencing prognosis. This should be phrased in a more cautious way.
Response: We have changed lines 378-381 to read: “Furthermore, the retrospective nature of the study design enabled us to leverage existing institutional data to explore potential relationships between patient characteristics and clinical outcomes, offering a deeper understanding of factors influencing snakebite prognosis at our institution.”
Reviewer 3 Report
Comments and Suggestions for Authors
Fascinating and well written paper. Thank you for the opportunity to review.
I have a few minor requested edits:
Reference 13 is not complete.
Table 3 – no p value for State, was this intended? There is a footnote for occupation.
It would be very interesting/useful to see the data for males/females/children time to arrival at hospital and total antivenom administration to understand whether these variables were related. You note this in the discussion, your paper would be stronger if you had this information. Particularly since sex, age, and hours between bite and hospitalization are in the simplified model. This may help elucidate why the XGBoost model had low sensitivity.
Your results are consistent with previous studies showing time between bite and treatment is an important indicator of outcome. You are deepening our understanding of this variable. Your work is important, although so is noting that it is consistent with prior studies. Please include this. A few options for your consideration follow. (There is no need to reference them all, and I am sure there are others you could choose)
Abdullahi A et al. Seasonal variation, treatment outcome, and its associated factors among the snakebite patients in Somali region, Ethiopia. Front Public Health. 2022;10:901414. 2022 Sep 27. doi:10.3389/fpubh.2022.901414.
Anderson VE et al. Early administration of Fab antivenom resulted in faster limb recovery in copperhead snake envenomation patients. Clin Toxicol (Phila). 2019;57(1):25-30. doi:10.1080/15563650.2018.1491982.
Gerardo CJ et al. Does This Patient Have a Severe Snake Envenomation?: The Rational Clinical Examination Systematic Review. JAMA Surg. 2019;154(4):346-354. doi:10.1001/jamasurg.2018.5069.
Gopalakrishnan M, et al. A simple mortality risk prediction score for viper envenoming in India (VENOMS): A model development and validation study. PLoS Negl Trop Dis. 2022;16(2):e0010183. Published 2022 Feb 22. doi:10.1371/journal.pntd.0010183.
Habib AG, Abubakar SB. Factors affecting snakebite mortality in north-eastern Nigeria. Int Health. 2011;3(1):50-55. doi:10.1016/j.inhe.2010.08.001.
Jayaraman T et al. Bite-to -needles time – an extrapolative indicator of repercussion in patients with snakebite. Indian J of Crit Care 2022;10.5005 10071-24344.
Narvencar K. Correlation between timing of ASV administration and complications in snake bites. J Assoc Physicians India. 2006;54:717-719.
Author Response
Fascinating and well written paper. Thank you for the opportunity to review.
Response: Thank you for your informative and helpful comments below. We have used your feedback to improve the quality of our paper.
I have a few minor requested edits:
Reference 13 is not complete.
Response: We have now completed this reference.
Table 3 – no p value for State, was this intended? There is a footnote for occupation.
Response: We have added a p-value for state as well, under Table 2.
It would be very interesting/useful to see the data for males/females/children time to arrival at hospital and total antivenom administration to understand whether these variables were related. You note this in the discussion, your paper would be stronger if you had this information. Particularly since sex, age, and hours between bite and hospitalization are in the simplified model. This may help elucidate why the XGBoost model had low sensitivity.
Response: We have added Figure S1 to illustrate the relationship between these variables.
Your results are consistent with previous studies showing time between bite and treatment is an important indicator of outcome. You are deepening our understanding of this variable. Your work is important, although so is noting that it is consistent with prior studies. Please include this. A few options for your consideration follow. (There is no need to reference them all, and I am sure there are others you could choose)
- Abdullahi A et al. Seasonal variation, treatment outcome, and its associated factors among the snakebite patients in Somali region, Ethiopia. Front Public Health. 2022;10:901414. 2022 Sep 27. doi:10.3389/fpubh.2022.901414.
- Anderson VE et al. Early administration of Fab antivenom resulted in faster limb recovery in copperhead snake envenomation patients. Clin Toxicol (Phila). 2019;57(1):25-30. doi:10.1080/15563650.2018.1491982.
- Gerardo CJ et al. Does This Patient Have a Severe Snake Envenomation?: The Rational Clinical Examination Systematic Review. JAMA Surg. 2019;154(4):346-354. doi:10.1001/jamasurg.2018.5069.
- Gopalakrishnan M, et al. A simple mortality risk prediction score for viper envenoming in India (VENOMS): A model development and validation study. PLoS Negl Trop Dis. 2022;16(2):e0010183. Published 2022 Feb 22. doi:10.1371/journal.pntd.0010183.
- Habib AG, Abubakar SB. Factors affecting snakebite mortality in north-eastern Nigeria. Int Health. 2011;3(1):50-55. doi:10.1016/j.inhe.2010.08.001.
- Jayaraman T et al. Bite-to -needles time – an extrapolative indicator of repercussion in patients with snakebite. Indian J of Crit Care 2022;10.5005 10071-24344.
- Narvencar K. Correlation between timing of ASV administration and complications in snake bites. J Assoc Physicians India. 2006;54:717-719.
Response: We have added most of these references in several places. Specifically, in lines 274-276: “...snakebite incidence peaked in April, at the beginning of the rainy season in this region, when farmers typically clear their farmlands; this is consistent with global and local studies [26–28].”, in line 332: “...early reception of antivenom is known to improve patient outcomes [38,39].”, and lines 363-364: “...in a similar manner to existing risk-prediction tools [44].”
Round 2
Reviewer 1 Report
Comments and Suggestions for Authors
I would like to thanks the authors for addressing my previous comments and for the revisions made to the manuscript. The changes have significantly improved the clarity and quality of the work. I have just one final suggestion regarding a specific paragraph that could benefit from further refinement:
On line 257 - 259, the authors writes: "random-forest models had higher specificity than sensitivity, meaning they were better at predicting which patients would experience amputation, debridement, or death, and worse at predicting who would recover."
This sentence misinterprets sensitivity and specificity:
- Sensitivity (true positive rate) refers to the model's ability to correctly identify patients who will experience amputation, debridement, or death.
- Specificity (true negative rate) refers to the model's ability to correctly identify patients who will not experience those outcomes (i.e., who will recover).
If the model has higher specificity than sensitivity, it means it is better at correctly identifying those who will recover (true negatives) and worse at identifying those who will experience severe outcomes (true positives).
Suggested Revision:
"Random-forest models had higher specificity than sensitivity, meaning they were better at identifying which patients would recover and worse at identifying who would experience amputation, debridement, or death."
Anton CAMACHO
Author Response
Thank you for pointing this out - we have revised those lines accordingly, as you have suggested.
Reviewer 2 Report
1. For the combined outcome, the authors should show the actual numbers for all three components. It is necessary to be able to interpret the outcome. e.g. 80 debridements 2 deaths and 9 amputations is different to 9 debridements 80 deaths and 2 amputations.
2. Age: In snakebite severity is determined by bodyweight. The age cut-off of 18 is meaningless and actually the cited reference 15 clearly states different age cut-offs and uses >12 as adolescent. This makes more sense to me. Further, the reference 34 is on pathophysiology of Echis in Rats and has nothing to do with the claimed „limited data on age“. Adding the median and IQR does not give a better description of the age range. I suggest grouping according to the classification the authors themselves suggest in Reference 15
3. Presence of clotting disorder: The authors state that 20MWBCT was done in all but do not present the result as influence variable on outcome. That is what I meant with my comment and it has not been addressed in my view.
Author Response
We appreciate the additional suggestions and have incorporated them into the revised version of our manuscript! Please find individual responses to Reviewer 2’s suggestions below.
1. For the combined outcome, the authors should show the actual numbers for all three components. It is necessary to be able to interpret the outcome. e.g. 80 debridements 2 deaths and 9 amputations is different to 9 debridements 80 deaths and 2 amputations.
Response: We have added the following sentence on line 205: “Of those that did not recover (n = 91, 9%), nine died, 78 experienced amputations, and four underwent debridement surgeries.“
2. Age: In snakebite severity is determined by bodyweight. The age cut-off of 18 is meaningless and actually the cited reference 15 clearly states different age cut-offs and uses >12 as adolescent. This makes more sense to me. Further, the reference 34 is on pathophysiology of Echis in Rats and has nothing to do with the claimed „limited data on age“. Adding the median and IQR does not give a better description of the age range. I suggest grouping according to the classification the authors themselves suggest in Reference 15
Response: Based on the categorisation in reference 15, we have defined new age categories, as noted in line 133: “those aged 0 to 11 (infant, toddler, childhood), aged 12 to 17 (adolescent), and aged 18 and over (adult).“ All analyses have been re-run with is new categorisation applied. Regarding your note about reference 34 - that was a typo on our part. We meant to direct you to reference 35: “Levine M, Ruha AM, Wolk B, Caravati M, Brent J, Campleman S. When It Comes to Snakebites, Kids Are Little Adults: a Comparison of Adults and Children with Rattlesnake Bites. Journal of Medical Toxicology [Internet]. 2020;(16):444–51. Available from: https://doi.org/10.1007/s13181-020-00776-6”
3. Presence of clotting disorder: The authors state that 20MWBCT was done in all but do not present the result as influence variable on outcome. That is what I meant with my comment and it has not been addressed in my view.
Response: We have added detail to line 391: “We were also limited by the breadth of existing data; for example, we were not able to include information on severity of swelling and presence/absence of clotting disorders, which are important factors that can influence patient outcomes. While it is part of routine practice, the 20-minute whole blood clotting test (20WBCT) is a simple, bedside test that can be used to detect coagulopathy in the context of snakebite envenoming and is only useful in resource-limited settings; it has limitations and is not a definitive diagnostic tool for all clotting disorders, and as such, is not an influential variable on outcome.”
Reviewer 3 Report
Comments and Suggestions for Authors
Thank you for the updated paper.
Author Response
Thank you for taking the time to suggest revisions and improve the quality of our paper!